# Enhancing Healthcare through Sensor-Enabled Digital Twins in Smart Environments: A Comprehensive Analysis

**DOI:** 10.3390/s24092793

**Published:** 2024-04-27

**Authors:** Sasan Adibi, Abbas Rajabifard, Davood Shojaei, Nilmini Wickramasinghe

**Affiliations:** 1School of Information Technology, Deakin University, Geelong, VIC 3220, Australia; 2School of Computing, Engineering and Mathematical Sciences, La Trobe University, Melbourne, VIC 3086, Australia; n.wickramasinghe@latrobe.edu.au; 3Centre for Spatial Data Infrastructures and Land Administration, Department of Infrastructure Engineering, The University of Melbourne, Parkville, VIC 3052, Australia; abbas.r@unimelb.edu.au (A.R.); shojaeid@unimelb.edu.au (D.S.)

**Keywords:** smart environments, smart hospitals, smart homes, digital twin, digital health, location-based services, Internet of Things (IoT), artificial intelligence (AI), wearable health devices

## Abstract

This comprehensive review investigates the transformative potential of sensor-driven digital twin technology in enhancing healthcare delivery within smart environments. We explore the integration of smart environments with sensor technologies, digital health capabilities, and location-based services, focusing on their impacts on healthcare objectives and outcomes. This work analyzes the foundational technologies, encompassing the Internet of Things (IoT), Internet of Medical Things (IoMT), machine learning (ML), and artificial intelligence (AI), that underpin the functionalities within smart environments. We also examine the unique characteristics of smart homes and smart hospitals, highlighting their potential to revolutionize healthcare delivery through remote patient monitoring, telemedicine, and real-time data sharing. The review presents a novel solution framework leveraging sensor-driven digital twins to address both healthcare needs and user requirements. This framework incorporates wearable health devices, AI-driven health analytics, and a proof-of-concept digital twin application. Furthermore, we explore the role of location-based services (LBS) in smart environments, emphasizing their potential to enhance personalized healthcare interventions and emergency response capabilities. By analyzing the technical advancements in sensor technologies and digital twin applications, this review contributes valuable insights to the evolving landscape of smart environments for healthcare. We identify the opportunities and challenges associated with this emerging field and highlight the need for further research to fully realize its potential to improve healthcare delivery and patient well-being.

## 1. Introduction

The advent of smart technologies in recent years has ushered in a transformative era, reshaping the very fabric of our daily lives, work environments, and healthcare landscapes. The convergence of smart home and smart building technologies has given rise to dynamic ecosystems that go beyond mere convenience, fundamentally altering our lifestyles and well-being. The driving forces behind the development of smart environments are multifaceted and will be covered in this article.

### 1.1. Introduction to Smart Environments

The rise of smart environments marks a paradigm shift in our interaction with the spaces that we inhabit. These intelligent ecosystems, encompassing smart homes, smart buildings, and smart hospitals, are seamlessly woven with cutting-edge technologies to create dynamic and responsive environments. At the core of this transformation lies the Internet of Things (IoT), enabling everyday objects to communicate and collect data, creating a rich fabric of information [1]. These data serve as the lifeblood of smart environments, powering various functionalities, from automated lighting and climate control in homes to real-time patient monitoring in hospitals.

Furthermore, advancements in artificial intelligence (AI) and machine learning (ML) empower smart environments with the ability to learn, adapt, and anticipate user needs. This translates to personalized experiences, enhanced comfort, and improved efficiency within these spaces. As sensor technologies continue to evolve, and the computing power increases, the capabilities and potential of smart environments are constantly expanding, paving the way for a future in which technology is seamlessly integrated with our lives to enhance well-being and productivity.

### 1.2. Context and Motivation

The integration of sensor-driven digital health systems within smart environments presents a transformative opportunity to revolutionize healthcare delivery. These systems leverage various sensors, such as wearables and in-home monitoring devices, to collect real-time data on individuals’ health parameters. This includes vitals like the heart rate, blood pressure, and sleep patterns, providing valuable insights into a user’s well-being. By integrating these data streams with digital twin technology, which creates virtual representations of physical systems, healthcare professionals can gain a holistic understanding of an individual’s health status and environment, which may offer the following opportunities.

**Enhanced real-time health monitoring**: Continuous data analysis allows for the early detection of potential health issues and timely intervention.**Personalized healthcare interventions**: Tailored treatment plans and preventative strategies can be developed based on individual health data and environmental factors.**Optimized resource allocation and operational efficiency**: Data-driven insights enable better resource allocation within healthcare systems, improving their efficiency and sustainability.**Improved patient experience**: Enhanced monitoring and personalized care can lead to better patient outcomes and satisfaction.

### 1.3. Main Research Question

This paper focuses on the following central research question:**To what extent and through what specific mechanisms can the integration of sensor-driven digital health systems with digital twin technology within smart environments contribute to improved healthcare delivery and patient outcomes?**

### 1.4. Novelty and Significance

This research explores the novel application of digital twin technology in healthcare settings, specifically focusing on its potential to


**Enhance real-time health monitoring and personalized interventions;**

**Optimize resource allocation and operational efficiency in healthcare systems in a secure manner;**

**Improve the patient experience and promote preventative healthcare strategies.**


### 1.5. Structure of the Paper

This paper follows the below structured approach to guide the reader through the intricacies of integrated smart environments.

Section 2: Foundational technologies—a review of the core technologies underpinning smart environments.Section 3: Literature review—research questions; methodologies; and the identification, critical evaluation, and synthesis of existing research on the integration of smart environments, digital health, and healthcare delivery.Section 4: Case studies: classical and contemporary smart environment systems—an extension to Section 3, presenting in-depth analyses of the real-world applications of smart environments and digital health.Section 5: Comparative analysis—highlighting the differences and similarities between smart environments (e.g., smart homes, buildings, hospitals).Section 6: Proposed solution—introducing a framework for the integration of digital health with digital twins in smart environments.Section 7: Impact analysis and future vision—exploring the potential impacts of smart environments on individuals and healthcare systems.Section 8: Challenges and future research directions—discussing the challenges and potential future research directions.Section 9: Conclusion—summarizing the key findings and future implications, followed by the references.

## 2. Foundational Technologies

The evolution of smart environments is underpinned by a set of foundational technologies. These technologies collectively redefine our living and working spaces, including the IoT, Internet of Medical Things (IoMT), AI, ML, and sensor networks.

*IoT and IoMT*—The Internet of Things forms the backbone of smart environments, enabling seamless communication between interconnected devices. In healthcare, the IoMT extends this connectivity to medical devices, facilitating remote monitoring and data collection. It encompasses a diverse range of medical devices and sensors that collect, transmit, and process medical data. IoMT devices can be implanted within the body (wearables), attached to the body (wearables), or embedded in the environment.

IoMT systems comprise medical sensors (e.g., smartwatches, wearables) that collect physiological data, communication modules (Bluetooth, Wi-Fi) for the transmission of data, and data processing units for initial analysis. Gateways act as intermediaries, potentially performing edge computing before relaying information to cloud platforms for storage, analysis, and visualization. Standardization (IEEE 1451, HL7, DICOM) ensures interoperability between devices, and while IoMT adoption is growing (wearables, blood pressure monitors), challenges persist in data security, privacy, and interoperability standards [1].

*AI and ML*—Artificial intelligence and machine learning bring a cognitive layer to smart environments. These technologies empower systems to learn and adapt, enhancing the efficiency and responsiveness of smart homes and buildings. Studies delving into practical implementations have demonstrated AI’s potential in optimizing energy consumption, improving security protocols, and personalizing user experiences within these intelligent spaces [2].

*Sensor Networks*—Often considered the sensory organs within the broader IoT framework, sensor networks play a pivotal role in data collection across diverse parameters, from temperature to occupancy. While sensor networks are integral to IoT systems, it is crucial to emphasize the interconnected nature of IoT devices, facilitated by these networks. The recent literature delves into the nuanced applications of sensor networks within real-world scenarios, elucidating their multifaceted roles. Studies have underscored their significance in environmental monitoring, healthcare diagnostics, and security management, emphasizing the collaborative functioning of the IoT devices interconnected through these networks [3]. Understanding sensor networks as an integral part of the IoT ecosystem is imperative, as they not only enable data collection but also foster seamless communication and coordination among interconnected devices. In essence, sensor networks serve as an essential component, interconnecting IoT devices to harness the collective intelligence of smart environments, contributing to enhanced functionalities in healthcare, security, and beyond [3].

### 2.1. Smart Environment Characteristics

The characteristics defining smart environments (e.g., homes, buildings, hospitals) extend far beyond conventional concepts. They offer a paradigm shift in our perceptions of living and working spaces. Smart environments exhibit distinctive features that transcend traditional notions of living and working spaces. In smart homes, automation and connectivity redefine comfort and security, while smart buildings integrate efficiency, sustainability, and technological advancements. The key features of smart environments include the following.

*Automation*—Smart environments leverage automation to dynamically adjust settings based on user preferences or environmental conditions. This can include lighting, temperature, and security systems, promoting comfort, convenience, and energy efficiency [4].

*Connectivity*—A defining feature is the ubiquitous connectivity facilitated by IoT and its healthcare extension (IoMT). This enables seamless communication and data exchange between devices, creating an intelligent ecosystem. This connected infrastructure fosters informed decision making, personalized experiences, and efficient automation [5].

*Efficiency and Sustainability*—Smart buildings, in particular, are characterized by a commitment to efficiency and sustainability. Integrating advanced technologies, these structures are designed to optimize resource usage, minimize environmental impacts, and enhance the overall operational efficiency. The integration of automation and connectivity within smart buildings contributes not only to a futuristic and streamlined user experience but also to a conscientious approach towards resource utilization and environmental stewardship [4,5].

*Focus on People*—While automation and efficiency are crucial, it is important to acknowledge the core human-centric purpose of smart environments. These spaces aim to enhance occupant well-being, comfort, and security through personalized automation and data-driven insights.

*Benefits and Applications*—Smart environments offer a wide range of benefits across various domains, as follows.

Healthcare: Sensor-driven systems can monitor health vitals, detect emergencies, and support independent living for people with chronic conditions.Energy Efficiency: Automation and real-time data analysis can optimize energy consumption and minimize environmental impacts.Safety and Security: Smart environments can proactively detect security threats, fires, or other hazards and provide real-time alerts.Convenience and Comfort: Automated adjustments based on user preferences or environmental conditions enhance comfort and convenience.

In essence, the distinctive features of smart environments, encompassing efficiency, connectivity, automation, and sustainability, mark a transformative era in the conception and realization of intelligent living and working spaces, including sensor-driven healthcare.

### 2.2. Features of Smart Healthcare and Hospital Systems

Technology enablers, based on user preferences or environmental conditions, foster comfort, convenience, and informed decision making, with the core purpose focusing on a human-centric perspective to enhance well-being and security and provide personalized experiences. These benefits extend across various domains, including healthcare (e.g., remote monitoring), energy efficiency, safety and security, and overall user comfort. The concept of smart hospital systems within smart buildings introduces innovations that reshape healthcare facilities. These innovations may be based on the incorporation of the following functionalities.

*Real-time Data Sharing*—The immediate sharing of patient data among healthcare professionals within the smart building, enhancing collaboration [6].

*Telemedicine Advancements*—The integration of telemedicine services within the hospital infrastructure, facilitating remote consultations and diagnostics [7].

*Automated Patient Care*—The implementation of automated systems for routine patient care tasks, optimizing healthcare workflows [8].

*Remote Patient Monitoring (RPM)*—RPM systems enable healthcare providers to remotely track patients’ health data, enhancing chronic disease management [9].

*Telepharmacy Systems*—Digital platforms that facilitate remote prescription management and pharmaceutical services [10].

*Teleconference Consultations*—Real-time virtual consultations between healthcare providers and patients, overcoming geographical barriers to healthcare access [11].

*Store-and-Forward Technology*—Enables the asynchronous sharing of medical data for collaboration among healthcare providers [12].

Such technological advancements have directly been transforming healthcare delivery, particularly for seniors, who require personalized and independent living solutions. Here, we explore key areas in which sensor-enabled technologies are playing a crucial role.

*Wearable Health Devices*—These non-invasive devices, such as smartwatches and fitness trackers, are revolutionizing preventive healthcare, through continuously monitoring vital signs and activity levels, providing valuable data for individuals and healthcare professionals. These data empower individuals to take a more proactive approach to their health, while enabling clinicians to identify potential health risks early on [13].

*Smart Home Health Monitoring*—These systems go beyond wearable devices, creating a comprehensive healthcare ecosystem within the home environment. Such integrated systems use various sensors to monitor a wider range of health parameters, providing a more holistic view of an individual’s well-being. These data can include sleep patterns, activity levels within the home, and environmental factors that may impact health. Smart home health monitoring systems offer significant benefits for both patients and caregivers, promoting independent living and enabling timely interventions when necessary [14].

*Medication Dispensers*—Automated systems that manage medication schedules, reducing the risk of errors [15].

*Telecare Systems*—Communication and alarm systems that enable remote monitoring and immediate responses in case of emergencies [16].

*Assistive Robotics*—Designed to help users with daily tasks and enable them to live independently [17].

The integration of digital health technologies into smart homes signifies a paradigm shift in healthcare delivery, including the following.

*Interconnected Health Ecosystems*—The seamless integration of health data from wearables and home monitoring systems into broader healthcare networks [18].

*AI-Driven Health Analytics*—Advanced analytics that process large datasets to derive meaningful insights for personalized healthcare [19].

### 2.3. Digital Twin Technology

Digital twin technology is a conceptual framework that creates a mirrored virtual representation of a physical system, such as an individual’s health profile or a hospital environment, aiding in predictive analysis and personalized interventions [20]. This process of creating a virtual replica is also referred to as digital twin mirroring. By continuously synchronizing the digital model with real-time data, the virtual representation closely resembles the physical system.

As a practical example, having a 3D model of a hospital and connecting IoT sensors to a digital twin platform allows the digital replica to be continuously synchronized with real-world data collected through various sensors and monitoring systems. Figure 1 (adapted from [21]) represents a schematic of the physical and digital twin integration. The core architecture of a digital twin typically involves three components [22] as follows.

*Physical Entity*—This refers to the real-world system (e.g., a patient, a hospital building) being mirrored.

*Digital Model*—This is a virtual representation of the physical entity, often built using 3D modeling software and incorporating relevant data points.

*Data Acquisition and Analytics*—Sensors and monitoring systems embedded within the physical entity continuously collect real-time data (e.g., vital signs, equipment readings, environmental factors). These data are then fed into the digital model, enabling its continuous updating and analysis.

Digital twin technology holds immense potential for personalized healthcare and improved healthcare delivery. The following are two healthcare-related applications.

*Patient Health Management*—By integrating data from wearable health devices, electronic health records (EHRs), and other sources, a patient’s digital twin can be created. This allows for comprehensive health monitoring, the early detection of potential issues, and personalized treatment plans.

*Hospital Optimization*—A digital twin of a hospital unlocks a new level of operational optimization. This virtual replica integrates 3D models of the physical space with real-time **sensor data** from medical equipment, environmental controls, and even (indirectly) patient locations. By analyzing this combined view, administrators can identify areas for improvement. This might involve underutilized operating rooms, inefficient patient flows leading to bottlenecks, or equipment nearing failure. Armed with these insights, they can make data-driven decisions to optimize resource allocation, minimize wait times for patients, and maximize the utilization of staff, equipment, and space. Ultimately, this translates into cost reductions, improved patient experiences, and potentially even better health outcomes through faster diagnoses and more efficient treatment delivery.

### 2.4. Location-Based Services (LBS)

Location-based services (LBS) utilize various technologies to determine a user’s location. For outdoors, global positioning systems (GPS) and cellular networks are common due to their high accuracy. For indoors, where GPS signals may be weak or unavailable, LBS relies on Wi-Fi triangulation, Bluetooth beacons, or other signal-based methods to provide a user’s location within a building or specific area.

*Indoor Positioning Systems (IPS)*—IPS typically focus on tracking individuals within a specific area. Their data can be integrated with a hospital’s digital twin to enhance patient safety and improve staff efficiency in locating patients for care or emergencies. Technologies that enable precise location tracking within buildings facilitate personalized services and navigation [23]. These technologies are mainly based on imaging, infrared, ultrasound, Wi-Fi, radio frequency identification (RFID), and Bluetooth. Figure 2 (adapted from [24]) depicts these technologies in terms of accuracy and coverage [24].

While LBS are not directly integrated into traditional smart homes, they play a vital role in broader healthcare delivery through utilizing GPS or similar technologies, which can be integrated with wearable health devices or medication management mobile applications to provide additional functionalities, such as the following.

*Emergency Response*—LBS can enhance emergency responses by automatically transmitting a user’s location when a fall or other critical event is detected by a wearable device.

*Medication Reminders*—LBS can trigger medication reminders based on a user’s location, ensuring adherence to medication schedules even when away from home.

*Geospatial Data*—Spatial data enable us to understand the environment and support applications such as asset tracking and emergency response systems [25].

### 2.5. Review of Comparative Foundational Technologies

Table 1 covers the key foundational technologies that are essential for integration with smart environments. It offers a comprehensive overview of their functionalities, strengths, and limitations. Cells with limited to no overlap between specific rows and specific columns (e.g., remote monitoring and foundational technologies, etc.) are marked with a hyphen (-) to indicate the lack of direct applicability.

Foundational technologies, including IoT, IoMT, AI, ML, and sensors, form the backbone of smart ecosystems, providing connectivity and automation. Smart home characteristics emphasize interconnected systems. Telehealth platforms introduce connectivity and real-time interactions, while home-based care technologies focus on user independence, real-time monitoring, and emergency responses. Digital health integration involves AI-driven analytics and interconnected health ecosystems. Smart hospital features showcase real-time data sharing and automated patient care. LBS leverage indoor positioning systems for swift emergency responses and personalized services. The comparison spans parameters such as *connectivity, automation, data analysis, remote monitoring, user independence, real-time interaction, emergency responses, integration complexity, privacy and security, interoperability, user-friendliness, scalability, cost-effectiveness, flexibility,* and *overall impacts*.

This section provides a foundation for an understanding of the multifaceted landscape of location-based integrated smart environments. The interplay of foundational technologies, the distinctive characteristics of smart homes and buildings, the transformative impacts of telehealth, and the integration of digital health technologies collectively define the evolution of intelligent environments.

The subsequent sections will build upon these insights, exploring the proposed solution, its impacts, challenges, and future avenues, including the targeted classical and contemporary smart environment systems covered in the following two sections.

## 3. Literature Review

This section delves into the key part of our research, employing a systematic literature review (SLR) to synthesize the existing knowledge regarding the integration of smart environments, digital health, smart hospitals, and LBS. However, before beginning the review process, we must first establish the research questions that guide our exploration [26].

### 3.1. Research Questions

As we navigate the burgeoning landscape of smart environments, where technology intertwines seamlessly with our daily lives, a myriad of questions arise regarding the integration of various technologies. This subsection aims to articulate the research questions that drive the exploration within this paper. These questions serve as a compass, guiding our inquiry into the secondary research with the multifaceted interplay of technological innovation, healthcare delivery, and spatial intelligence.

*Main Research Question*—“To what extent, and through what specific mechanisms, does the integration of smart home and smart building technologies with digital health capabilities, smart hospital innovations, and location-based services contribute to the creation of intelligent living and working environments that demonstrably improve well-being and productivity?”

This central question forms the core of our investigation, prompting a deeper exploration into the functionalities and impacts of integrated smart environments.

*Secondary Research Questions*—The main research question is further complemented by a series of secondary research questions that delve into the characterization of the enabling technologies and their applications within smart environments. These secondary questions provide a more granular perspective on the functionalities and limitations.

**Foundational Technologies**: How effectively do combinations of foundational technologies (IoT, IoMT, AI, ML, sensor networks) enable and enhance key functionalities (comfort, security, efficiency) in smart environments, and what are the limitations of the current implementations?**Smart Home and Building Characteristics**: How do automation and connectivity features in smart homes and buildings demonstrably improve user comfort, security, and efficiency?**Telehealth Impact**: How do telehealth interventions (remote monitoring, consultations, medication management) demonstrably improve healthcare accessibility, delivery, and patient outcomes in different contexts (chronic conditions, remote areas)?**Home-Based Care Technologies**: How effectively do home-based care technologies (wearables, monitoring systems, assistive robots) support personalized and independent living for seniors, and what are the ethical and privacy considerations to address?**Digital Health Integration**: How do digital health technologies (interconnected ecosystems, AI analytics, digital twins) demonstrably improve healthcare delivery and patient outcomes within smart home environments, and what are the challenges and opportunities for wider adoption?**Smart Hospital Features**: How do smart hospital features (real-time data sharing, telemedicine, automated care) demonstrably improve operational efficiency, patient care, and overall healthcare experiences, and what are the challenges and potential unintended consequences?**Location-Based Services**: How do location-based services (indoor positioning, geospatial data) demonstrably enhance the capabilities of smart environments by facilitating personalized services, asset tracking, and emergency responses, and what are the privacy and security concerns to address?

These secondary research questions provide a roadmap for the investigation of the intricacies, challenges, and potential solutions within this dynamic intersection of technology and well-being.

### 3.2. Selection and Analysis

Building upon the established research questions, the SLR methodology ensures a rigorous and transparent approach to gathering and analyzing relevant information. This process involves several key steps.

*Search Strategy*—Our search strategy was designed to capture a broad spectrum of literature addressing the integration of smart home and smart building technologies with digital health capabilities, smart hospital innovations, and LBS. Key databases such as IEEE Xplore, PubMed, Scopus, and Web of Science were searched using a combination of keywords related to “*smart environments*”, “*digital twins*”, “*healthcare*”, “*IoT*”, “*AI*”, and “*sensor networks*”. The search was limited to articles published in the last decade to focus on the most recent technological advancements and their implications.

*Inclusion and Exclusion Criteria*—The inclusion criteria were as follows:Studies focusing on the integration of smart technologies with healthcare applications;Articles detailing the use of IoT, IoMT, AI, ML, and sensor networks in smart environments;Research highlighting the impact of smart technologies on well-being and productivity within healthcare settings.

The exclusion criteria included the following:
Studies lacking empirical evidence or those primarily theoretical in nature;Outdated publications with low relevance to the research questions.

*Data Extraction and Coding*—Data relevant to the research questions were extracted from the selected studies. These data included key findings, methodologies, and outcome measures. The coding scheme refers to the process of categorizing and organizing the information extracted from the selected research articles for further analysis.

*Critical Evaluation*—The quality and methodological rigor of the selected studies were critically assessed. This step helped to evaluate the trustworthiness and generalizability of the findings.

The selection process involved two-stage screening: an initial title and abstract review to identify potentially relevant articles, followed by a full-text review to confirm their eligibility based on the inclusion criteria. This dual-step approach ensured the inclusion of pertinent literature while maintaining a high standard of relevance and quality.

Our analysis employed both qualitative and quantitative methodologies. The SLR facilitated a structured and comprehensive exploration, identifying key themes, methodologies, and findings across the selected studies. Meanwhile, the meta-analysis offered a statistical lens to assess the effectiveness and impact of smart technologies in healthcare environments, enabling us to quantify the trends, patterns, and overall significance.

Data relevant to the research questions were also extracted from the selected studies. These data included key findings, methodologies, and outcome measures. The mentioned coding scheme was then used to categorize and organize the extracted data for further analysis.

### 3.3. Methodologies

A thorough literature review serves as the foundation for the synthesis of existing knowledge and extraction of valuable insights. Various methodologies are employed in the current literature to comprehensively understand and analyze the intricate facets of integrated technologies. In this section, we explore several research methodologies documented in the present literature.

*Systematic Literature Review (SLR)*—The SLR is a systematic approach to reviewing the literature that follows a predefined protocol and set of criteria to identify, select, compare and contrast, and evaluate relevant studies [26]. The comprehensive approach includes integrating systematic review protocols with a meta-analysis to analyze and synthesize the findings from selected studies. This methodology ensures a rigorous and transparent approach to gathering and analyzing the relevant literature.

*Meta-Analysis*—A meta-analysis is a statistical technique for the combination and analysis of data from multiple studies to draw overarching conclusions [27]. While this was not the main focus in this particular study, it is, however, a valuable tool for future research endeavors seeking to quantitatively assess the impact of integrated technologies.

*Content Analysis*—This is a method for the systematic analysis of the content of textual, visual, or audio data to identify patterns, themes, and trends [28]. Content analysis can be a useful tool in understanding the user experience within smart environments.

*Surveys and Questionnaires*—Structured surveys or questionnaires are used to gather opinions, perceptions, and experiences from experts, practitioners, or end-users [29].

Table 2 compares and contrasts the above secondary research methodologies, including their strengths and weaknesses.

*Selection of Methodology for Data Collection, Analysis, and Synthesis*—To achieve a comprehensive analysis of the existing research, a combined methodology using an SLR (as the main) and meta-analysis (as the secondary approach) was adopted. The SLR method emphasizes the structured analysis of research. It involves systematic searches using predefined criteria to select relevant studies in specific areas like smart environments and digital health. This approach ensures transparency and allows for the replication of the research process. Complementarily, a meta-analysis adds a quantitative layer, analyzing the data from the chosen studies to identify trends, patterns, and statistical significance, which quantifies the impact of integrated technologies. This amalgamation of an SLR and meta-analysis allows for a detailed and quantified understanding of the technologies’ integration, enhancing the review’s robustness and providing a comprehensive view based on secondary data.

### 3.4. Research Findings

The synthesis of the selected studies revealed several core areas in which smart environments intersect with healthcare enhancement [1,2,3,4,5,6,7,8,9,10,11,12,13,14,15,16,17,18,19,20,21,22,23,24,25,26].

*Foundational Technologies*—The effective deployment of IoT, IoMT, AI, ML, and sensor networks is critical in creating smart environments that improve healthcare delivery and patient outcomes. These technologies underpin the development of intelligent systems that enhance comfort, security, and efficiency in both living and healthcare settings.

*Impact on Healthcare Delivery*—Digital health technologies, including telehealth, wearables, and home-based monitoring systems, have shown significant promise in improving healthcare accessibility, delivery, and outcomes. Smart hospitals, leveraging real-time data sharing and automated care, demonstrate operational efficiency and enhanced patient care.

*Ethical and Privacy Considerations*—The integration of smart technologies in healthcare settings raises important ethical and privacy concerns. Addressing these issues is crucial in ensuring patient trust and safeguarding personal data.

*Contemporary* vs. *Classical Smart Environments*—The evolution from classical to contemporary smart environments highlights a shift towards more integrated, intelligent systems. Contemporary smart environments utilize advanced AI, ML, and sensor networks to offer personalized, anticipatory services and health monitoring capabilities.

*Challenges and Opportunities*—While the integration of digital health technologies within smart environments presents numerous opportunities to enhance healthcare delivery, challenges remain in terms of adoption, interoperability, and the management of ethical considerations, which will be further discussed in the final sections.

In summary, this SLR underscores the transformative potential of integrating smart environments with healthcare technologies. By leveraging digital twins and sensor-enabled systems, there is a considerable opportunity to enhance well-being and productivity. However, addressing the challenges of technology integration and managing ethical considerations are paramount in realizing this potential.

## 4. Case Studies: Classical and Contemporary Smart Environment Systems

This section is an extension to Section 3 (literature review), providing an in-depth analysis of classical and contemporary smart environment systems through carefully selected case studies. Each case study has been chosen based on the key research questions, criteria that ensure relevance, innovation, and the potential to highlight the transformative impact of digital technologies in healthcare and smart environments. Our focus is to bridge the gap between theoretical discussions and practical implementations.

### 4.1. Case Selection Criteria

This section’s case studies showcase the practical application of foundational technologies. This includes IoT, AI, and ML, all crucial for smart environments. The focus is on the applicability to the digital twin technology, demonstrating its real-world use within these environments. The criteria included the novelty of the technology, its impact on healthcare delivery and patient experiences, and its relevance to the challenges and opportunities within smart healthcare systems. To ensure a comprehensive analysis, case studies were selected based on the following criteria.

*Technology Integration*—The case studies showcase how various smart environment technologies (e.g., sensors, wearables, location-based services) are combined to create intelligent systems.

*Focus on Healthcare*—The cases prioritize applications that demonstrate the potential of smart environments to enhance healthcare delivery, patient well-being, or remote health monitoring.

*Balance between Classical and Contemporary Systems*—The selection includes established (classical) smart environment systems alongside more recent advancements (contemporary) to illustrate the evolution of the field.

### 4.2. Classical and Contemporary Smart Environment Systems

Smart environments have evolved significantly from classical systems to contemporary ones, driven by advancements in technology. Classical environments laid the groundwork with early home automation systems (HAS) like X10, enabling basic functionalities such as remote lighting control [30]. Building management systems (BMS) focused on the centralized control of building infrastructure for energy efficiency and security, with early examples including programmable thermostats (e.g., the Nest Learning Thermostat) [31,32]. Contemporary smart environments leverage the convergence of technologies like IoT for seamless control. AI and ML personalize user experiences with virtual assistants like Alexa [33]. Sensor networks capture real-time data for various purposes, like occupancy detection and air quality measurement in smart buildings [34]. The integration of digital health technologies like smartwatches and home-based monitoring devices contributes to preventative healthcare [35]. Smart hospitals utilize EHR, telemedicine, and robotic surgery systems to improve healthcare delivery [36]. LBS with geospatial intelligence provide features like smart navigation and emergency response systems [37]. These advancements highlight the vast capabilities of contemporary smart environments compared to their classical counterparts, which will be discussed in more detail in the proceeding subsections.

### 4.3. Case Study Analysis

This section delves into a series of case studies [38] that exemplify the evolution and effectiveness of smart environments in healthcare, guided by the integration of foundational technologies and digital twin concepts, starting with the classical systems used for smart environments.

*Smart Home Integration: Nest Learning Thermostat*—The Nest Learning Thermostat exemplifies a classical smart home technology that leverages **sensor data** and machine learning to optimize energy use and user comfort. The Nest integrates with various smart home devices, creating an automated system that learns the user’s preferences and adjusts the heating and cooling settings accordingly. This case study demonstrates the potential of smart environments to enhance the energy efficiency within residences, potentially contributing to cost savings and environmental benefits [32,39]. The next subsections will include a number of contemporary systems used for smart environments.

*Apple Watch: AI-Driven Health Monitoring*—The Apple Watch represents a significant advancement in AI-based wearable health devices, offering functionalities that extend beyond fitness tracking to include heart rate monitoring, electrocardiography (ECG), and fall detection. This case study exemplifies how wearable technology can play a crucial role in preventive healthcare and emergency situations, showcasing the direct application of sensor-driven healthcare systems in everyday life. It illustrates the seamless integration of health monitoring technologies into users’ lives, underscoring the potential for wearables to enhance patient care and health outcomes [40].

*Cleveland Clinic Abu Dhabi: Smart Hospital Implementation*—The Cleveland Clinic Abu Dhabi’s adoption of smart hospital features, such as real-time data sharing, telemedicine advancements, and automated patient care, highlights the potential of digital twin technology and AI in transforming healthcare delivery. This case study is presented to address the lack of depth in discussing smart healthcare applications, providing a comprehensive overview of how digital twins can optimize hospital operations, patient flows, and clinical outcomes [41].

*Amazon Go: Location-Based Services in Retail*—Amazon Go’s utilization of LBS through indoor positioning systems (based on sensor fusion, computer vision, and deep learning) offers insights into the broader applications of LBS beyond healthcare, particularly in enhancing consumer experiences. While not directly tied to healthcare, this case study is included to demonstrate the versatility and potential of LBS in smart environments, offering a basis for the consideration of its implications in healthcare settings, such as navigating large hospital complexes or enhancing patient and asset tracking systems [42].

*The Edge, Amsterdam: Smart Building Sustainability*—The Edge in Amsterdam is recognized for its smart building sustainability, incorporating energy-efficient technologies and IoT to create one of the world’s greenest office spaces. This case relates to the broader discourse on smart environments by illustrating the role of smart technologies in achieving sustainability goals. It underscores the potential for smart systems to significantly reduce energy consumption and carbon footprints, aligning with the growing emphasis on environmental sustainability in healthcare facilities [43].

These case studies [44,45,46,47,48] have been analyzed to highlight their contributions to advancing smart environment technologies and their implications for healthcare delivery. By showcasing these practical implementations, this section aims to bridge theoretical discussions with real-world applications, providing a comprehensive understanding of the transformative potential of digital technologies in smart environments (Table 3).

## 5. Comparative Analysis

In an effort to provide a more nuanced understanding of the evolution and potential of smart environments, this section incorporates a detailed comparative analysis framework. This framework not only distinguishes between classical and contemporary smart environments but also emphasizes the transformative role of digital twin technology in bridging the gap between these two paradigms. Through this lens, we explore key dimensions that highlight their functionalities, strengths, limitations, and the emergent possibilities enabled by digital twin integration. The key dimensions for differentiation between classical and contemporary systems include the following.

*Connectivity*—A foundational element in our comparative analysis, connectivity has evolved from the wired confines of classical smart environments to embrace the wireless, interconnected fabric of the modern IoT ecosystem. This evolution marks a shift towards ubiquitous connectivity, enabling devices to communicate and collaborate more effectively. Contemporary smart environments, augmented by digital twin technology, further advance this connectivity, allowing for the creation of virtual replicas that mirror physical devices in real time. This dual-layered interaction enhances intelligent decision making and scenario simulation [49].

*Automation*—Traditional automation was characterized by static commands within classical systems. Today, the integration of AI and machine learning in contemporary environments introduces dynamic, adaptive automation. Digital twins enrich this landscape by enabling a predictive and personalized approach to automation, learning from user interactions and preferences to anticipate needs and adjust environments accordingly [50].

*Data Analysis*—The leap from basic data processing to advanced analytics powered by AI and ML has been significant. Digital twin technology amplifies this capability, offering a comprehensive view by synchronizing real-time data from physical and virtual counterparts. This enables deeper insights, predictive analytics, and optimization strategies, thereby enhancing the intelligence and efficiency of smart environments [51].

*Remote Monitoring*—Where classical systems offered limited remote capabilities, contemporary solutions thrive on extensive, real-time monitoring facilitated by IoT and digital health innovations. Digital twins extend this functionality, providing a holistic view of health and environmental conditions, thereby revolutionizing patient care and monitoring with unprecedented accuracy and immediacy [52].

*User Independence*—The transition from limited user autonomy to personalized, adaptive systems represents a critical shift towards user-centered design. Digital twins play a pivotal role in this evolution, enabling environments that not only adapt to individual preferences but also evolve with them, thus promoting greater independence and satisfaction [53].

*Real-Time Interaction*—The capability for seamless real-time interactions, especially in healthcare and smart homes, has been significantly enhanced by contemporary smart systems. Digital twins enhance these interactions by offering real-time data synchronization and simulation, thereby improving decision making, efficiency, and user experiences [54].

*Emergency Response*—Enhanced emergency response capabilities are a hallmark of contemporary smart environments. Through the integration of location-based services and IoT, coupled with digital twin technology, these systems can provide more intelligent, swift responses to emergencies, ensuring enhanced safety and well-being [55].

*Integration Complexity and Interoperability*—While contemporary systems have improved interoperability, the integration of diverse devices and platforms remains complex. Digital twins can mitigate these challenges by serving as a unifying layer that facilitates seamless integration and communication across various devices and systems, highlighting the importance of standardized integration approaches [56].

*Privacy and Security*—As the sophistication of smart environments increases, so does the complexity of privacy and security concerns. Digital twin technology introduces additional layers of data, necessitating robust security measures and privacy protocols to protect sensitive information in both physical and virtual spaces [57].

*User-Friendliness*—The advancement in user interface design has significantly improved the accessibility and convenience of contemporary smart environments. Digital twins contribute to this improvement by offering intuitive, real-time visualizations of physical and virtual interactions, thus enhancing user engagement and adoption [58].

*Scalability*—The ability to accommodate an expanding network of devices and users is crucial. Digital twins support scalable solutions in smart environments by providing flexible frameworks that adapt to growing demands and changing user needs, ensuring sustainability and adaptability [59].

*Cost-Effectiveness*—Contemporary smart environments offer a spectrum of solutions ranging from affordable to high-end. Digital twins add value by optimizing resource use and operational efficiency, potentially lowering costs over time and offering scalable solutions that cater to a broad range of user needs and financial capacities [60].

*Flexibility and Adaptability*—The agility of contemporary systems to adapt to diverse user needs is a defining feature. Digital twins enhance this flexibility, allowing for rapid adjustments and customization based on real-time data and user feedback, ensuring that smart environments remain relevant and responsive to evolving requirements [61].

By integrating digital twin technology into the fabric of smart environments, we unlock a new dimension of connectivity, intelligence, and user-centric design. This comparative analysis underscores the pivotal role of digital twins in transitioning from classical to contemporary smart environments, highlighting their potential to revolutionize the ways in which we live, work, and interact with our surroundings. The overall impact of smart environments is transformative in contemporary settings, reshaping living and working spaces and redefining healthcare delivery. This represents an evolutionary leap from the incremental impact of classical systems on lifestyle and efficiency [62].

In conclusion, as summarized in Table 4, this comparative analysis illuminates the progression from classical to contemporary smart environments, highlighting their respective strengths, weaknesses, and profound impacts on the ways in which we live, work, and experience healthcare.

## 6. Proposed Solution: Enhancing Smart Environments with Digital Twin Technology

In response to the challenges unveiled in our comparative analysis of classical and contemporary smart systems, we present a comprehensive solution designed to elevate the functionality, security, and user experience within smart environments. Our proposed solution revolves around the implementation of the Intelligent Connectivity Framework (ICF), which is the combination of the foundational technologies (e.g., IoT, AI), BMS, and HAS, and addresses the limitations identified in both classical and contemporary smart systems.

*Addressing Challenges in Smart Environments*—Our comparative analysis (Section 5) revealed limitations in both classical and contemporary smart environments. These limitations encompass restricted connectivity, limited automation capabilities, and data security concerns. To address these shortcomings, we propose a comprehensive solution, the *Intelligent Connectivity Framework (ICF)*, and its enhanced novel version (E-ICF).

### 6.1. Intelligent Connectivity Framework (ICF)

The ICF is a conceptual approach that offers solutions to the limitations of healthcare systems. It combines core technologies (IoT, AI) with existing infrastructure (BMS, HAS, cyber-physical systems (CPS)) to create a secure and user-friendly environment [44,45,46]. This framework (Figure 3) fosters seamless integration, real-time data exchange, and improved interoperability across healthcare devices and systems. This ultimately enhances decision making, optimizes resource allocation, and improves patient outcomes [63].

*Optimized Connectivity*—The ICF ensures seamless communication between devices through advanced protocols, fostering real-time data exchange and eliminating connectivity bottlenecks [64].

*Adaptive Automation*—The framework employs AI algorithms to enable the system to adapt its behavior based on user preferences and real-time **sensor data**. This personalization enhances the user experience and caters to individual needs [65].

*Enhanced Security*—Robust security protocols are implemented within the ICF to safeguard data privacy and system integrity, addressing security concerns in smart environments [66].

### 6.2. Enhanced Intelligent Connectivity Framework (E-ICF)

An enhanced version of the ICF (E-ICF) may incorporate digital twin technology. This digital replica mirrors the physical smart environment, enabling real-time data analysis and user interaction simulation (Table 5). The E-ICF unlocks several functionalities that address the limitations identified in the previous section.

*Predictive Maintenance*—By analyzing **sensor data** and historical information, the digital twin can anticipate potential equipment malfunctions or inefficiencies, allowing for preventive maintenance and minimizing downtime [67].

*Personalized Recommendations*—The E-ICF leverages user data and health readings to generate personalized recommendations for health management, comfort optimization, and resource allocation. This focus on user experience is crucial [68].

*Resource Optimization*—The E-ICF analyzes real-time and historical data to optimize energy consumption and resource allocation within the smart environment [69].

### 6.3. Human Experience in E-ICF Design

The E-ICF inherently incorporates user-centric elements as follows [70].

*Adapting to User Preferences*—The E-ICF tailors its automation and recommendations based on user input and historical data, enhancing user comfort and satisfaction.

*Prioritizing User Security*—Robust security protocols within the E-ICF safeguard user data and privacy, fostering trust and promoting user adoption.

*Facilitating Personalized Interactions*—The E-ICF allows for user input and feedback, enabling continuous improvement and user-centered development.

### 6.4. Input and Output Parameters

For seamless integration, the E-ICF considers various input and output parameters.


*Input Parameters:*
System configuration data;User preferences;Security protocols;Real-time **sensor data** (temperature, energy consumption, occupancy);User interaction data (appliance usage, health readings, activity patterns);External environmental data (weather, traffic, air quality).


These inputs define the data requirements and communication security levels.


*Output Parameters:*
**Optimized Connectivity and Automation**: The E-ICF ensures seamless device communication and adaptive automation based on user preferences and real-time data.**Predictive Maintenance**: The digital twin anticipates potential issues and triggers preventive measures.**Personalized Health Management**: The E-ICF analyzes health data to provide proactive health insights and recommendations.**Resource Optimization**: Energy use and resource allocation are optimized based on real-time data and predictive analysis.**Enhanced Security**: Security protocols are informed by real-time threat detection and anomaly analysis within the digital twin.


These outputs enable adaptive automation aligned with user behavior and preferences, while robust security measures ensure data privacy and system integrity.

### 6.5. Steps towards Proof-of-Concept and Implementation

The E-ICF’s design positions it as a central layer, facilitating communication between devices within smart environments. It integrates advanced AI algorithms to enable adaptive automation that learns from user interactions and adjusts system behavior accordingly. Stringent security protocols are embedded at both the hardware and software levels to fortify the protection against cyber threats.

The proposed implementation plan for the E-ICF consists of five phases.

*System Audit and Data Integration*: Assess the existing infrastructure and establish a data flow between the physical and digital environments.*Digital Twin Model Development*:Create a high-fidelity digital replica of the smart environment using **sensor data** and historical information.This replica should encompass relevant details based on the specific use case (e.g., home environment vs. hospital setting) [71].*AI Algorithm Integration*: This phase focuses on integrating AI algorithms within the digital twin for simulation, prediction, and optimization functionalities. Below is a breakdown of the process.**Algorithm Selection**: Identify and select appropriate AI algorithms based on specific tasks. For instance, predictive and proactive maintenance might utilize anomaly detection algorithms, while personalized recommendations could leverage machine learning for pattern recognition.**Training and Validation**: Train the chosen algorithms using historical data and sensor readings from the physical environment. Rigorous validation ensures that the algorithms perform accurately and avoid generating biased outputs.**Continuous Learning**: The E-ICF should incorporate mechanisms for continuous learning. This allows the AI algorithms to adapt to evolving user behavior and environmental changes over time, enhancing the effectiveness of the system.*User Trials and Feedback*: User trials are crucial in evaluating the E-ICF’s usability, effectiveness, and user experience. The following describes how to conduct them.**Recruiting Participants**: Select a representative group of users who reflect the target audience for the smart environment (e.g., homeowners, healthcare professionals).**Scenario Development**: Develop realistic scenarios that simulate everyday use cases within the smart environment. This allows for the testing of various functionalities and user interactions.**Data Collection**: Gather qualitative and quantitative data during the user trials. Qualitative data, such as user feedback and observations, provide insights into the user experience and satisfaction. Quantitative data, collected through sensor readings and system logs, help to measure performance metrics like efficiency and accuracy.**Iterative Improvement**: Analyze the data from the user trials to identify areas for improvement. Refine the E-ICF based on the user feedback and address any usability issues encountered.*Scalable and Interoperable Integration*: This phase ensures that the E-ICF integrates seamlessly with diverse smart environments and future technologies.**Standardized Interfaces**: Develop standardized interfaces for the E-ICF to facilitate communication with various devices and systems from different manufacturers. This promotes interoperability and avoids vendor lock-in.**Modular Design**: Design the E-ICF with a modular architecture. This allows for easy customization and future expansion to accommodate new functionalities and technologies.**Scalability Testing**: Conduct scalability testing to ensure that the E-ICF can handle an increasing number of devices and users within a smart environment. This is crucial for real-world deployment in large-scale environments.

### 6.6. Steps towards Feasibility Study

A comprehensive feasibility study will verify the viability of the E-ICF solution. It will systematically address the challenges identified in the comparative analysis, focusing on enhancing connectivity, automation, and security. The evolving technological landscape, particularly advancements in AI, IoT, and cybersecurity, aligns well with the proposed solution.

The resource requirements for the E-ICF’s implementation encompass a multidisciplinary approach.

*Addressing Identified Challenges*—Proactive maintenance, personalized health management, and resource optimization address the limitations of existing systems.

*Alignment with Technology Advancements*—Leverages advanced AI, IoT, and cybersecurity technologies for robust implementation.

*Resource Requirements*—Similar to the base ICF solution, requiring AI experts, cybersecurity specialists, and system integrators, along with additional expertise in digital twin development.

*Financial Investment*—Requires additional resources for digital twin model development and AI algorithm implementation.

A summary of the proposed solution is provided in Table 6.

### 6.7. Limitations and Future Research Directions

While the E-ICF framework holds promise in addressing the limitations in smart environments, acknowledging its potential limitations is crucial.

*Data Privacy and Security Challenges*—Implementing robust security measures and ensuring user data privacy throughout the system life cycle remains an ongoing challenge.

*Computational Requirements*—Running complex AI algorithms and maintaining a high-fidelity DT may require significant computational resources, potentially limiting the scalability of the E-ICF in resource-constrained environments.

*Ethical Considerations*—The extensive data collection and analysis inherent in the E-ICF raise ethical concerns regarding user consent, data ownership, and potential biases in AI algorithms.

Considering the review nature of this article, further research needs to articulate, expand upon, and implement the suggested basic steps described for the ICF and E-ICF and to address these limitations and explore the below directions.

*Development of lightweight AI algorithms*—Optimizing AI algorithms for efficient processing on resource-constrained devices.

*Federated learning techniques*—Enabling collaborative learning across multiple devices while preserving data privacy.

*Standardized ethical frameworks*—Developing clear guidelines for data collection, usage, and responsible AI development in smart environments.

## 7. Impact Analysis and Future Vision

The integration of smart environments, digital health, and location-based services has ushered in transformative changes, profoundly impacting individuals, healthcare systems, and businesses.


*Individuals:*
**Enhanced quality of life**: Smart homes offer personalized automation and optimization for comfort and convenience (e.g., temperature control, lighting adjustments).**Proactive health management**: Wearables and home health monitoring enable the early detection of health issues and preventive care.**Increased independence**: Assistive robotics and smart home technologies empower individuals, especially the elderly, to maintain self-sufficiency.



*Healthcare Sector:*
**Improved patient care efficiency**: Smart hospitals leverage real-time data sharing and telemedicine for faster diagnosis, treatment, and remote monitoring.**Enhanced chronic disease management**: Digital health integration reduces hospital visits and facilitates better disease management.**Personalized healthcare interventions**: AI-powered analytics and digital twin technology enable predictive analysis for personalized care.


*Businesses*:**Operational efficiency**: Smart buildings promote energy savings, sustainability, and streamlined operations, leading to cost reductions.**Telehealth platforms**: Businesses in the healthcare domain can expand their reach and offer new services.**Personalized healthcare interventions**: Retail businesses can leverage location-based services for targeted marketing, increasing customer engagement and loyalty.

### 7.1. Quantifying the Impact: Case Studies

Case studies demonstrate the tangible benefits of this integration.

*Smart homes*—Automated energy management can lead to a 30% reduction in energy bills [72]. Real-time health monitoring enables the early detection of health issues and timely intervention [73].

*Smart hospitals*—Real-time data sharing reduces diagnosis and treatment times by 20% [74]. Automated patient care systems can increase the overall operational efficiency by 15% [75].

*Retail businesses*—Indoor positioning systems for personalized marketing can lead to a 20% increase in customer engagement [76]. Emergency services leveraging geospatial data for swift responses can reduce the emergency response times by 25% [77].

These case studies collectively showcase improved efficiency and convenience for individuals, marking an evolution in healthcare and business practices in response to technological advancements. The real-world impact extends beyond mere convenience, contributing to a paradigm shift in how we live, receive healthcare, and conduct business.

*Beyond Convenience: A Paradigm Shift*—While convenience is certainly a benefit, the true impact goes far deeper. This technological revolution is fundamentally reshaping how we live, receive healthcare, and conduct business. 

### 7.2. Future Vision: Digital Twin for Intelligent and Responsive Environments

Building upon the proposed E-ICF framework, the integration of digital twin technology paves the way for a future with truly intelligent and responsive environments. These smart spaces will seamlessly adapt to our needs and proactively contribute to our well-being, efficiency, and sustainability.

### 7.3. Summary of Inputs and Outputs of the Digital-Twin-Ready Model


**Inputs:**
**Sensor data:** Temperature, humidity, energy consumption, occupancy, air quality, noise levels, lighting conditions, appliance usage, etc.**User interaction data:** Preferences, activity patterns, health readings, feedback on comfort and energy use, etc.**External data:** Weather forecasts, traffic updates, air quality reports, security threats, etc.**Historical data:** Past trends, system performance data, maintenance records.



**Outputs:**
**Real-time insights:** Current state of the environment, potential issues, energy consumption patterns, comfort levels, etc.**Predictive analysis:** Anticipated maintenance needs, resource optimization suggestions, personalized health recommendations, etc.**Adaptive automation commands:** Adjustments to temperature, lighting, ventilation, security settings, etc., based on real-time data and predictions.**Personalized recommendations:** Suggestions for improved comfort, health, energy efficiency, and overall well-being.**Security alerts:** Detecting potential threats and vulnerabilities within the smart environment.


This enhanced solution (Figure 4), with its digital twin capabilities, paves the way for a future in which smart environments are seamlessly integrated with our lives.

## 8. Challenges and Future Research Directions

The integration of smart environments, digital health, and location-based services, while demonstrably transformative, faces significant challenges. Recognizing and addressing these obstacles is crucial in crafting effective solutions and guiding future research efforts.

### 8.1. Key Challenges

*Interoperability*—The lack of standardized protocols and open architectures hinders seamless communication between the diverse devices and systems within smart environments. Addressing this necessitates industry-wide collaboration towards standardization and the creation of open frameworks. Research efforts should explore the development and adoption of common communication protocols to facilitate smooth data exchange.

*Privacy and Security*—Smart environments collect and manage sensitive user data. Striking a balance between innovation and robust privacy measures is paramount. Ongoing research and development are required to create and implement advanced cybersecurity protocols that ensure data protection. This includes exploring privacy-preserving technologies such as anonymization and differential privacy.

*Integration Complexity*—Integrating new technologies into existing smart systems can be complex, demanding substantial infrastructure modifications. Streamlining integration processes is critical for wider adoption. Research should focus on developing modular and adaptable architectures that simplify integration and reduce the reliance on large-scale infrastructure upgrades.

*User-Centric Design*—Despite the advanced functionalities, user-friendliness remains a persistent challenge. Bridging the gap between technological capabilities and user expectations requires a paradigm shift towards user-centric design principles. This includes conducting user experience studies to understand user needs and preferences and incorporating intuitive interfaces and clear user feedback mechanisms into smart environments.

*Scalability and Cost-Effectiveness*—Scalability and cost remain hurdles, particularly in deploying smart environments in diverse socio-economic contexts. Developing cost-effective solutions and scalable deployment strategies is essential for widespread adoption and societal impacts. Research should explore cost-efficient technologies and innovative deployment models suited for various socio-economic settings.

### 8.2. Future Research Directions

Looking towards the future, several research directions can be pursued to overcome these challenges and further advance smart environments.

*Standardization and Interoperability*—Develop and promote standardized protocols and open frameworks to ensure seamless communication between diverse devices and systems.

*Privacy-Preserving Technologies*—Explore and integrate privacy-preserving technologies like anonymization and differential privacy to mitigate security risks and enhance user trust.

*Simplified Integration Methods*—Develop modular and adaptable architectures to simplify the integration of new technologies into existing smart systems.

*Human-Centered Design Principles*—Conduct user experience studies to understand user needs and preferences and prioritize user-centric design principles in developing smart environments.

*Scalable Deployment Strategies*—Explore cost-efficient technologies and innovative deployment models to facilitate widespread adoption in diverse contexts.

*Integration of Emerging Technologies*—Investigate the integration of emerging technologies such as AI, ML, and blockchain into smart health environments for enhanced functionalities and data security.

*Ethical and Social Implications*—Conduct a thorough examination of the ethical and social implications of the widespread adoption of smart environments, addressing potential biases and ensuring responsible development and deployment.

### 8.3. Summary

As the smart environment landscape continues to evolve, addressing these challenges and pursuing the outlined research directions will be critical for the continued refinement of intelligent living and working spaces. Achieving smarter, more efficient, and user-centric environments requires interdisciplinary collaboration, a steadfast commitment to tackling complex challenges, and a focus on ethical and responsible innovation.

## 9. Conclusions

The comparative analysis presented in this paper has provided critical insights into the concurrent evolution of smart home and building technologies, their integration with digital health capabilities, and the pivotal role of location-based services. The key findings highlight the foundational technologies underpinning smart environments, the transformative impact on healthcare delivery, the fusion of digital health and smart environments, and the significance of location-based services in creating intelligent ecosystems.

### 9.1. Addressing Research Questions

As outlined in Section 3 (Secondary Research Key Questions), this research explores the multifaceted landscape of smart environments, digital health capabilities, smart hospitals, and location-based services.

*Main Research Question*—In the current literature, the integration of smart home and smart building technologies with digital health capabilities, smart hospital innovations, and location-based services has been identified as a multifaceted process. The literature reveals that smart environments leverage foundational technologies, including IoT, IoMT, AI, ML, and sensor networks, to shape and optimize living and working spaces. The amalgamation of these technologies contributes to the creation of intelligent environments that enhance well-being and productivity.


*Secondary Research Questions:*
**Foundational Technologies**—These technologies (IoT, IoMT, AI, ML, sensor networks) act as the backbone, enabling seamless communication, data collection, and adaptive learning for enhanced efficiency within smart environments.**Smart Home and Smart Building Characteristics**—Automation and connectivity redefine comfort, security, and efficiency. Automated systems respond to user preferences, adjusting the settings to create intelligent living spaces.**Telehealth**—Telehealth services like remote patient monitoring, consultations, and medication delivery significantly impact healthcare accessibility. The literature discusses how telehealth bridges gaps, improving overall patient outcomes.**Home-Based Care Technologies**—Wearable health devices, smart monitoring systems, medication dispensers, and assistive robotics address the need for independent living, particularly for seniors. These technologies contribute to promoting autonomy and well-being.**Digital Health Integration**—Integrating digital health technologies, including interconnected ecosystems, AI-driven health analytics, and digital twin technology, reshapes healthcare delivery within smart homes.○*Digital Twin Technology*—This technology creates a virtual replica of the physical space, enabling real-time analysis, predictive maintenance, and personalized recommendations. For instance, in smart homes, digital twins can optimize energy consumption, anticipate equipment failures, and suggest personalized health interventions. This reinforces the potential of smart environments to proactively care for our well-being.○*Enhanced Intelligent Connectivity Framework (E-ICF)*—This proposed novel solution builds upon existing functionalities and leverages digital twin technology to create a more intelligent and adaptive environment. By integrating AI, advanced automation, and robust security protocols, the E-ICF addresses challenges like interoperability and user privacy. Its focus on predictive maintenance, personalized health management, and resource optimization aligns with the evolving needs of smart environments.**Smart Hospital Features**—Real-time data sharing, telemedicine advancements, and automated patient care redefine healthcare facilities. The literature explores how smart hospital features improve the operational efficiency, patient care, and the overall healthcare experience.**Location-Based Services**—*Indoor positioning systems (IPS) and geospatial data enhance the capabilities of smart environments. The literature discusses their applications in facilitating personalized services, asset tracking, and swift emergency responses*.


These responses collectively provide insights into the intricate relationships and advancements in smart environments, addressing the core research questions and paving the way for a comprehensive understanding of the technology’s integration and its impacts on well-being.

### 9.2. Key Insights and Future Implications

This analysis offers several key insights and highlights areas for future research and development.

*Foundational technologies* like IoT, IoMT, AI, and ML form the backbone of smart environments, demanding continued research to optimize their functionalities.

*Smart homes and buildings* redefine living and working spaces, requiring the further exploration of user needs and preferences to personalize experiences.

*Digital health integration* offers a paradigm shift in healthcare delivery, but necessitates further research on privacy-preserving techniques and ethical considerations surrounding data ownership and algorithmic bias.

*Smart hospitals* showcase advancements in healthcare facilities, with opportunities for further research on optimizing data-driven decision making and resource allocation.

*Location-based services* enhance the capabilities of smart environments, but require further exploration to ensure responsible data collection and utilization.

*Digital twin technology* holds immense promise for proactive and personalized smart environments. However, careful consideration is needed regarding the ethical implications and data privacy concerns. Robust governance frameworks and user control mechanisms are essential to ensure responsible adoption.

The proposed *E-ICF framework* offers a potential solution by prioritizing user needs, employing privacy-preserving techniques, and adhering to open standards and ethical AI practices. By fostering transparency, communication, and user-centric design, the E-ICF can navigate digital twins’ ethical landscape, ensuring trust to pave the way for smart environments that enrich our lives while upholding responsible and sustainable practices.

### 9.3. Beyond Technological Advancements: A Holistic Approach

This research goes beyond the technology lens, providing a holistic view that considers ethical, societal, and user-centric concerns. Despite the complexities, the potential for smart, adaptable environments to improve our lives remains immense.

### 9.4. Shaping the Future: A Collective Endeavor

The journey towards smart environments is a collective endeavor requiring interdisciplinary collaboration, ethical considerations, and a commitment to shaping a future in which technology enriches the human experience. Further research must explore the ethical, societal, and user-centric aspects alongside technological advancements.

This review highlights the possibilities afforded by healthcare digital transformation [78]. This transformation strives to ensure that individuals can experience optimal health and well-being across various settings, promoting high-quality lifestyles. It further emphasizes the comfort and assurance of high-quality acute care when necessary. Reference [78] highlights Health 4.0 as one of the key factors in the digital transformation of healthcare delivery. It leverages technologies such as AI, big data, and the IoT/IoMT to create a more interconnected, data-driven healthcare system. This approach aims to improve patient outcomes, optimize resource allocation, and enhance data-driven decision making and the overall personalized healthcare experience across diverse smart environments [78].

### 9.5. Digital Twins: A Promising Future with Responsible Implementation

While digital twin technology promises a future characterized by proactive and personalized smart environments, its adoption necessitates the careful consideration of ethical implications and data privacy concerns. Questions surrounding data ownership, algorithmic bias, and transparency in decision making demand robust governance frameworks and user control mechanisms.

The E-ICF offers a responsible development approach through

Prioritizing user needs through features like adaptable automation and personalized recommendations based on user preferences;Employing privacy-preserving techniques to safeguard user data and build trust within the system;Adhering to open standards and ethical AI practices to ensure transparency and responsible development.

By fostering transparency, communication, and user-centric design, the E-ICF can navigate the ethical landscape of digital twins. This ensures trust and paves the way for smart environments that enrich our lives while upholding responsible and sustainable practices.

### 9.6. Final Thoughts: A Vision for the Future

Smart environments represent more than merely a technological evolution—they embody a profound shift in how we inhabit and interact with our living spaces. The symbiotic relationship between technology, healthcare, and spatial intelligence holds the promise of not only responsive living spaces but active contributors to our well-being and productivity.

In conclusion, this paper comprehensively analyzes the evolving landscape of smart environments, emphasizing the synergy between smart home/building technologies, digital health advancements, and location-based services. The exploration highlights the role of foundational technologies in shaping intelligent spaces and the transformative impact on healthcare delivery.

The proposed E-ICF framework, which leverages digital twin technology, offers a promising solution to address the challenges in smart environments. However, responsible implementation requires acknowledging the ethical considerations and prioritizing user privacy.

As we move forward, collective effort is crucial to ensure that technological advancements prioritize the well-being of individuals and societies. Further research, interdisciplinary collaboration, and a commitment to ethical considerations are essential in shaping a future in which smart environments enrich human lives in a responsible and sustainable manner.

## Figures and Tables

**Figure 1 sensors-24-02793-f001:**
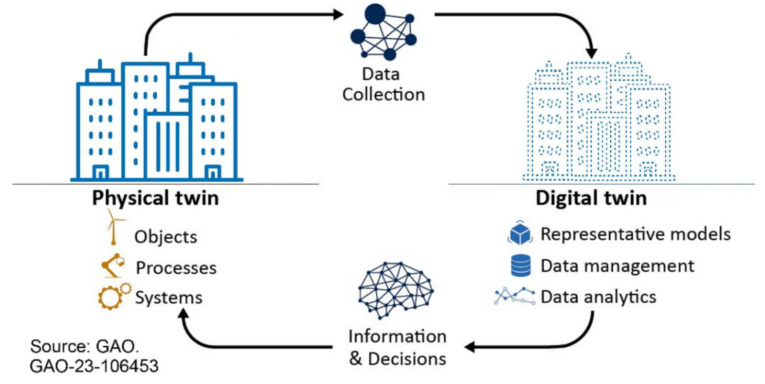
Physical and digital twin integration schematic (adapted from [21]).

**Figure 2 sensors-24-02793-f002:**
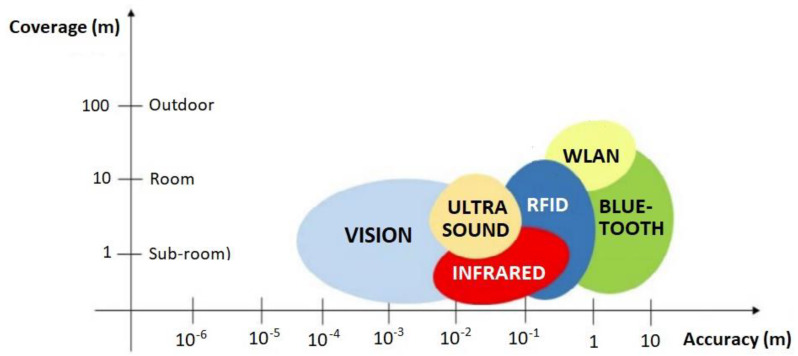
An overview of indoor positioning technologies and their accuracy and coverage (adapted from [24]).

**Figure 3 sensors-24-02793-f003:**
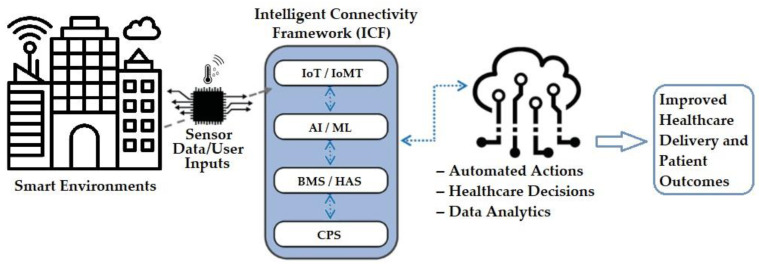
The conceptual Intelligent Connectivity Framework (ICF).

**Figure 4 sensors-24-02793-f004:**
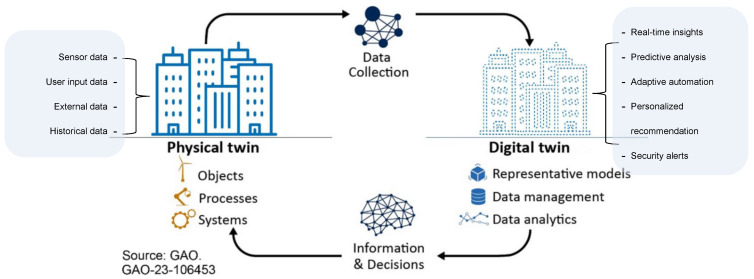
Digital-twin-ready model inputs and outputs (adapted from [21]).

**Table 1 sensors-24-02793-t001:** Review of comparative foundational technologies.

Parameters	Foundational Technologies	Smart Home Characteristics	Telehealth	Home-Based Care Technologies	Digital Health Integration	Smart Hospital Features	Location-Based Services
**Connectivity**	IoT, IoMT, AI, ML, Sensors	Interconnected Ecosystems	Platform Connectivity	Interconnected Devices	Interconnected Health Ecosystems	Immediate Data Sharing	Indoor Positioning Systems
**Automation**	AI, ML	Automated Systems	-	Automation of Health Monitoring	-	Automated Patient Care	-
**Data Analysis**	AI, ML	-	AI-Driven Analytics	AI-Driven Health Analytics	AI-Driven Health Analytics	-	-
**Remote Monitoring**	-	Enabled by Sensors and Automation	RPM Systems	Wearable Health Devices	-	Real-Time Data Sharing	-
**User Independence**	-	Enabled by Automation and Assistive Technologies	-	Assistive Technology	-	-	-
**Real-Time Interaction**	-	-	Teleconference Consultations	Teleconference Consultations	-	Telemedicine Advancements	-
**Emergency Response**	-	-	Swift Emergency Responses	Telecare Systems	-	-	Swift Emergency Responses
**Integration Complexity**	High	Medium	High	Medium	High	Medium	Medium
**Privacy and Security**	Technology-Dependable Protocols	Security Protocols in Place	Compliance Standards	Privacy Measures	Security Measures	Stringent Security Measures	Privacy Considerations
**Interoperability**	Challenging	Interoperable Systems	Interoperable Platforms	Interoperability Challenges	Interconnected Health Systems	-	Interoperable Technologies
**User-Friendliness**	Easy-to-Use Interfaces	User-Friendly Interfaces	User-Friendly Platforms	Intuitive Interfaces	User-Friendly Integrations	-	User-Friendly Applications
**Scalability**	Scalable Systems	Scalable Systems	Scalable Platforms	Scalability in Monitoring Devices	Scalable Health Ecosystems	Scalable Hospital Infrastructures	Scalable Services
**Cost-Effectiveness**	Consideration of Each Technology	Cost-Effective Solutions	Cost-Effective Platforms	Cost-Effective Devices	Cost-Effective Integrations	-	Cost-Effective Implementations
**Flexibility and Adaptability**	Flexibility in Implementation	Adaptive Automation Systems	Adaptive Platforms	Adaptive Health Devices	Flexible Integrations	-	Adaptive Positioning Systems
**Overall Impact**	Transformative	Significant Impact	Transformative	Enhancing Independent Living	Paradigm Shift in Healthcare	Reshaping Healthcare Delivery	Enhancing Intelligent Environments

**Table 2 sensors-24-02793-t002:** Comparison of secondary research methodologies.

Methodology	Description	Key Characteristics	Strengths	Limitations
Systematic Literature Review (SLR)	Systematic approach to reviewing the literature with a predefined protocol and criteria for the identification, selection, and evaluation of relevant studies [23].	- Rigorous and systematic process - Predefined protocol and criteria - Comprehensive overview of existing literature	- Provides comprehensive synthesis of existing knowledge - Establishes foundation for further research	- Time-consuming process - Potential for bias in study selection
Meta-Analysis	Statistical technique for combination and analysis of data from multiple studies to draw overarching conclusions [24].	- Statistical synthesis of data - Draws overarching conclusions - Quantitative approach	- Enhances statistical power and generalizability - Identifies trends across studies	- Requires homogenous data for meaningful analysis - Sensitivity to data quality and study heterogeneity
Content Analysis	Systematic analysis of textual, visual, or audio data to identify patterns, themes, and trends [25].	- Identifies patterns and trends - Applicable to textual, visual, or audio data - In-depth qualitative insights	- Useful in exploring in-depth qualitative aspects - Flexible and adaptable to different types of data	- Subjectivity in coding and interpretation - Resource-intensive process
Case Study Analysis	In-depth examination of specific cases or instances to gain insights into the application, challenges, and outcomes of integrated technologies [26].	- In-depth exploration of specific cases - Insights into real-world applications - Focus on challenges and outcomes	- Provides rich, context-specific insights - Allows for exploration of complex phenomena	- Limited generalizability - Susceptible to researcher bias and interpretation
Surveys and Questionnaires	Use of structured surveys or questionnaires to gather opinions, perceptions, and experiences from experts, practitioners, or end-users [27].	- Gathering opinions and perceptions - Structured data collection - Insights from experts or end-users	- Efficient in collecting large amounts of data - Facilitates standardized data collection	- Response bias and variability in participant responses - Limited depth compared to qualitative methods

**Table 3 sensors-24-02793-t003:** Key findings and comparative analysis of classical and contemporary smart environments.

Aspect	Classical Smart Environment	Contemporary Smart Environment
**Overview**	Foundational stage shaping smart homes and buildings.	Paradigm shift driven by technology, connectivity, and AI.
**Early Home Automation Systems**	Basic functionalities: lighting, heating, and security control.	IoT integration for seamless control and monitoring.
**Basic Building Management Systems (BMS)**	Centralized control of HVAC, lighting, and security.	Advanced BMS with real-time analytics, optimizing efficiency.
**IoT Integration**	Limited connectivity and automation.	Proliferation of IoT devices for interconnected, responsive ecosystems.
**AI and Machine Learning**	Absence of AI and ML capabilities.	AI and ML for adaptive, personalized services.
**Sensor Networks and Real-Time Analytics**	Basic sensor integration with limited data capture.	Sophisticated sensor networks and real-time analytics for informed decision making.
**Digital Health Integration**	Absence of digital health technologies.	Integration of wearables, health monitoring, and AI-driven health analytics.
**Smart Hospital Innovations**	Limited technology integration in healthcare facilities.	Smart hospitals with real-time data sharing, telemedicine, and automated care.
**Location-Based Services and Geospatial**	Limited precision in location-based services.	Advanced indoor positioning and precise geospatial data for enhanced experiences.

**Table 4 sensors-24-02793-t004:** Summary of characteristics of classical and contemporary smart environments.

Dimension	Classical Smart Environment	Contemporary Smart Environment
**Connectivity**	Reliance on wired systems with limited inter-device communication	Extensive use of wireless technologies, especially IoT, facilitating seamless connectivity and communication between devices, including digital twin mirroring
**Automation**	Basic automation systems with predefined commands	Advanced automation driven by AI and machine learning, adapting to user behavior and preferences
**Data Analysis**	Limited data processing capabilities	Utilizing AI and ML for sophisticated data analysis, enabled by digital twin synchronization for predictive insights and optimizations
**Remote Monitoring**	Minimal remote monitoring capabilities	Remote monitoring powered by IoT and digital health technologies, enhanced by digital twin for holistic view
**User Independence**	Limited user autonomy and customization	Empowerment of users through personalized, adaptive systems, promoting independence
**Real-Time Interaction**	Restricted real-time interactions	Seamless real-time interactions, especially in healthcare and smart home settings
**Emergency Response**	Basic emergency response features	Swift and intelligent emergency responses, facilitated by location-based services and IoT
**Integration Complexity**	High integration complexity and limited interoperability	Improved interoperability, but with increased complexity due to the diversity of devices and systems
**Privacy and Security**	Limited security protocols	Strong security tackles privacy concerns, but new challenges emerge with the ever-growing number of connected devices
**User-Friendliness**	Basic interfaces with limited user-friendliness	Intuitive interfaces designed for user convenience and ease of interaction
**Scalability**	Limited scalability	Scalable systems that can accommodate a growing number of devices and users
**Cost-Effectiveness**	Consideration of cost, often limited by technology constraints	Varied cost-effectiveness, with a broader range of affordable options and high-end solutions
**Flexibility and Adaptability**	Limited flexibility in system adaptation	Adaptive systems catering to diverse user needs and preferences
**Overall Impact**	Incremental impact on lifestyle and efficiency	Transformative impact, reshaping living and working spaces and redefining healthcare delivery

**Table 5 sensors-24-02793-t005:** Comparisons between Intelligent Connectivity Framework (ICF) and Enhanced ICF (E-ICF).

Feature	ICF	E-ICF
Digital twin	No	Yes
Predictive maintenance	No	Yes
Personalized health management	No	Yes
Resource optimization	Basic	Advanced (through digital twin insights)
AI capabilities	Basic	Advanced (real-time simulations, etc.)

**Table 6 sensors-24-02793-t006:** Summary of proposed solution.

Aspect	Details
**Solution Overview**	- Introduction of the Enhanced Intelligent Connectivity Framework (E-ICF), incorporating digital twin technology for a holistic, adaptive, and secure smart environment solution.
**Input Parameters**	- System configuration data. - User preferences for customization. - Security protocols. - Real-time **sensor data** from IoT devices (temperature, energy consumption, occupancy). - User interaction data (appliance usage, health readings, activity patterns). - External environmental data (weather, traffic, air quality).
**Output Parameters**	- *Optimized Connectivity and Automation:* Seamless communication and adaptive behavior based on user preferences, real-time data, and AI-driven insights. - *Predictive Maintenance:* Anticipation of potential malfunctions or inefficiencies through digital twin analysis, triggering preventive measures. - *Personalized Health Management:* Proactive health insights and recommendations based on individual health data analyzed by the digital twin. - *Resource Optimization:* Energy consumption and resource allocation optimized based on real-time data, predictive analysis, and digital twin insights. - *Enhanced Security:* Security protocols informed by real-time threat detection and anomaly analysis within the digital twin.
**Proof-of-Concept**	- The E-ICF acts as a bridge between physical and digital environments, facilitating two-way data flow. - Real-time data are fed into the digital twin for analysis and the generation of insights. - Digital twin sends optimization commands to the physical environment. - Advanced AI algorithms within the E-ICF enable real-time simulations, issue identification, and personalized recommendations.
**Implementation Plan** **(5 Phases)**	1. *System audit and data integration*: Assess the existing infrastructure and establish a data flow between the physical and digital environments. 2. *Digital twin model development*: Create a high-fidelity digital replica using **sensor data** and historical information. 3. *AI algorithm integration*: Implement AI algorithms for simulation, prediction, and optimization within the digital twin. 4. *User trials and feedback*: Conduct user trials to refine the solution and gather feedback. 5. *Scalable and interoperable integration*: Ensure seamless integration with diverse smart environments and future technologies.
**Feasibility Study**	- Addresses identified challenges through proactive maintenance, personalized health management, resource optimization, and AI-driven insights. - Leverages advanced AI, IoT, cybersecurity, and digital twin technologies. - Requires similar human resources (AI experts, cybersecurity specialists, system integrators) to the base ICF solution, with additional expertise in digital twin development. - Financial investment needed for digital twin model development and AI algorithm implementation.
**Conclusion**	- The E-ICF transcends the limitations of existing systems, creating a truly intelligent and responsive environment. - It fosters a future in which smart spaces proactively contribute to well-being, efficiency, and sustainability. - The feasibility study confirms the alignment with the current technological capabilities and demands.

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
