# Peer review of "Enhancing Healthcare through Sensor-Enabled Digital Twins in Smart Environments: A Comprehensive Analysis"

_sensors, 2024, doi:10.3390/s24092793_

Round 1

Reviewer 1 Report

Comments and Suggestions for Authors

no comment 

Comments on the Quality of English Language

really, it is very good writing 

Author Response

Dear reviewer,

The authors thank you for the time and effort in reviewing the manuscript. The paper has undergone a major revision and the thorough list of changes have been documented, which is attached to this message.

With regards,

Sasan Adibi,

The corresponding author

Reviewer 2 Report

Comments and Suggestions for Authors

1. The style of piling up in this article is too obvious. A sentence is a paragraph. The whole article doesn't look like an article, but just a framework. Many topics are not discussed academically.

2. In the title of this paper, the Digital twin appears. However, in the text, the substantive issues of Digital twin are not described, for example, its technological architecture, business architecture, application cased and advantages, implementation platform and so on. Undoubtedly, the discussion about DT is superficial, especially only three related research documents are cited, which can't reflect the current research status.

3. The research motivation does not revolve around the gaps in academic research and the difficulties in practice. In the pages 9, the authors put forward the so called Side Secondary Research Questions. What are the research status about these questions? Or, are these questions brand-new research guides? Seemly, this manuscript is more like an entry-level report of research, rather than a comprehensive analysis.

4. Whether it is smart home, smart environment or smart medical care, it should be human-centered. However, this report obviously circumvents this core issue. Human intelligence and artificial intelligence, the interaction and symbiosis between physical space and cyberspace are also not mentioned. This greatly reduces the height and significance of this study.

5. Wearable devices and home robots are becoming more and more popular, and their role in intelligent man-machine environment is becoming more and more important. The manuscript also does not introduce their application summary and research progress in depth. There are only three articles about wearable devices. Technically, this manuscript has no academic value.

6. The AI technology in smart home and smart healthcare is also becoming popular. The manuscript also does not introduce its research progress, practice, specific cases and development problems.

It is suggested that the author carry out extensive research, carefully concise academic issues and present a high-level revised version.

Author Response

(The authors gave the same response as above.)

Reviewer 3 Report

Comments and Suggestions for Authors

Thank you for providing the opportunity to review this insightful work and offer feedback to further enhance this contribution.

I found the paper to cover a very relevant area since the integration of digital technologies within smart environments has emerged as a transformative paradigm, particularly in the realm of healthcare services and supply.

In detail, I see some very positive aspects in the paper, but also areas for improvement that should be undertaken.

The paper offers a comprehensive analysis of the integration of smart home technologies with digital health capabilities, smart hospital innovations, and location-based services. This thorough exploration provides valuable insights into the evolving landscape of healthcare and technology integration. I also think the recap of key points effectively summarizes the foundational technologies, transformative impacts, and potential solutions discussed throughout the paper. It serves as a useful reference for readers to grasp the main contributions of the research.

The implications section provides valuable insights into the future development and integration of smart environments, emphasizing the importance of continued research and responsible adoption. Additionally, the discussion on ethical implications and data privacy concerns demonstrates a thoughtful consideration of societal implications. I found the introduction of Digital Twin technology and the Enhanced Intelligent Connectivity Framework / E-ICF as potential solutions to be innovative. These frameworks address challenges such as interoperability, privacy, and user-centric design, thereby enhancing the practical relevance of the research.

However, there are some critical aspects and space for improvements that can be undertaken.

The paper covers a wide range of subtopics and topics related to technology enablers, smart environments and healthcare. To improve clarity and depth, focus on a few key topics and provide a more in-depth analysis rather than briefly touching on numerous aspects. To make it clear, technologies, methods and concepts are sometimes only explained in a few sentences. Each individual technological or methodological component is worth examining against the background of the research question. It might be helpful to narrow things down and delve into individual aspects rather than writing a very broad, general paper.

While the paper offers valuable insights, the presentation could be streamlined for better readability. Consider organizing the content more logically, perhaps by grouping related themes together or following a clearer structure that guides the reader through the analysis seamlessly. It also would be beneficial to foster the section discussing the limitations of the research. Addressing potential biases, gaps in the literature, or methodological constraints would provide a more balanced perspective and enhance the credibility of the findings. In addition to discussing implications, consider offering more practical recommendations for stakeholders involved in implementing digital twin technology and E-ICF in healthcare settings. This could include guidance on governance frameworks, data management strategies, and user engagement approaches to ensure successful deployment and adoption.

Author Response

(The authors gave the same response as above.)

Reviewer 4 Report

Comments and Suggestions for Authors

This paper delves into the transformative potential of integrating smart technologies with digital health capabilities, smart hospital innovations, and location-based services to create intelligent living and working environments. By leveraging foundational technologies like IoT, IoMT, AI, and ML, the aim is to enhance well-being, productivity, and healthcare delivery in smart environments. The following are suggestions from the reviewer:

1.          The abstract broadly covers the manuscript's themes but misses detailing key findings and innovations, which are crucial for highlighting its unique contributions. I suggest refining the abstract to clearly summarize the most critical insights or breakthroughs, like novel applications of digital twin technology in healthcare, to enhance its informativeness and reflect the paper's novelty.

2.          The abstract mentions exploring smart technologies and a solution framework but overlooks the review's methodological rigor. Incorporating a brief description of the methodology, including literature selection criteria and analysis techniques, would enhance the abstract's value by providing insight into the review's methodological foundation.

3.          The introduction provides a wide-ranging overview of smart technologies but could benefit from a more focused approach. Specifying the central thesis or research question early on would sharpen the focus and better engage readers by clearly stating the research's challenges and intended impact.

4.          The objectives in the introduction are introduced later than ideal, which may confuse readers about the study's goals. Bringing the statement of objectives to the end of the first or second paragraph would offer a clearer guide for the audience, ensuring they understand the study's direction and aims from the start.

5.          While the paper's structure is mentioned at the introduction's end, mentioning the methodological approach earlier would solidify the study's foundation. A quick overview of the selection process for technologies and applications, inclusion criteria, and the analytical framework would highlight the research's rigor and aid in understanding the inquiry's methodology, enhancing the study's credibility and clarity for readers.

6.          Simplifying the use of acronyms and technical jargon with concise explanations upon their first mention would make the section more accessible to a broader audience, improving comprehension for those less familiar with the topic.

7.          The evaluation of user experience in smart environments appears to be underexplored. Including user experience studies, such as usability testing and satisfaction surveys, would provide valuable insights into the efficacy of these technologies from the user's perspective.

8.          The methodology section could more clearly articulate the study's limitations and potential biases. Outlining these limitations would enhance the transparency of the research process and contribute to the scholarly integrity of the paper.

This paper offers an insightful exploration into the integration of smart technologies and digital health within intelligent environments, underpinning significant potential for enhancing life and healthcare quality. However, to elevate its contribution to the academic discourse, substantial revisions are advised. Key areas for improvement include refining the abstract to succinctly highlight innovations, incorporating a detailed methodology early on for rigor transparency, and providing a clearer focus within the introduction. Additionally, the paper would benefit from a more accessible presentation of technical information, a deeper examination of user experience, and a thorough discussion of the study's limitations. Given these considerations, I recommend reconsideration of the manuscript after major revisions are made, ensuring the paper fully captures its intended impact and relevance in the evolving field of smart technologies.

Comments on the Quality of English Language

Moderate editing of English language required

Author Response

(The authors gave the same response as above.)

Round 2

Reviewer 2 Report

Comments and Suggestions for Authors

1. In the Section 2, there are three aspects: IoT and IoMT ,Sensor Networks,AI and ML.The authors noted that while sensor networks are integral to IoT systems, it's crucial to emphasize the interconnected nature of IoT devices facilitated by these networks. But the problems such as what are the IoMT, and what are the basic components, system architecture, standards, real status quo and availability of IoMT, are still not clear.

2.  Optimization can include maximizing or minimizing certain variables or parameters to achieve the best outcome. In the line 259, please further confirm what are the contents of Hospital Optimization. What you have talked are mainly data collections, monitoring, rather than optimization.

3. Table 1 lacks a head caption.

4. A framework is a basic structure underlying a system, concept, or project that serves as a foundation for development. But in the Subsection 6.1, I cannot see a picture to illustrate the basic domains, I/O relations, information flowing of the ICF. It’s better present it by a diagram.

5. In the Subsection 6.5, there is no real proof of concept. There is no example, no experiment, no prototype, so you did not prove your concept. You only give a basic steps, so it is still in your concept.

6. A feasibility study is an analysis that examines the practicality and viability of a proposed project or system. It aims to assess whether the project is technically feasible, economically viable, and operationally achievable. In the Subsection 6.6, obviously, I cannot any conclusions of the so called feasibility. What you have discussed are also the contents or basic requirements.

7. The contents of the Subsection 7.3 is better to combine with the Figure 1.

8. In the Figure 4, what is Health 4.0? Please give a clear definition.

Despite the author's efforts to revise the original manuscript, many arguments are still very simple. Therefore, I recommend this manuscript still needs major revision.

Comments on the Quality of English Language

Good

Author Response

Dear Reviewer,

The authors thank you for the time for the second review.

Here are the comments/suggestions and the corrections incorporated in the revised paper:

  1. In the Section 2, there are three aspects: IoT and IoMT ,Sensor Networks,AI and ML.The authors noted that while sensor networks are integral to IoT systems, it's crucial to emphasize the interconnected nature of IoT devices facilitated by these networks. But the problems such as what are the IoMT, and what are the basic components, system architecture, standards, real status quo and availability of IoMT, are still not clear.

Thank you for your valuable feedback. I have addressed your comments by incorporating a dedicated paragraph on the Internet of Medical Things (IoMT). This paragraph provides a clear definition of IoMT, along with its basic components and relevant standards. I believe this additional information strengthens the foundation for understanding the role of IoMT in smart environments for healthcare.

  1. Optimization can include maximizing or minimizing certain variables or parameters to achieve the best outcome. In the line 259, please further confirm what are the contents of Hospital Optimization. What you have talked are mainly data collections, monitoring, rather than optimization.

Thank you for your feedback on the "Hospital Optimization" subsection. You've rightly pointed out that the original description focused more on data collection and monitoring rather than optimization.

To address this, we have revised and extended the paragraph to emphasize how the digital twin facilitates proactive decision-making. The new version highlights how real-time data analysis helps identify areas for improvement, such as underutilized resources or bottlenecks in patient flow. This focus on optimization aligns with your suggestion and strengthens the overall message of the subsection.

  1. Table 1 lacks a head caption.

Not sure what it meant by “head caption”, but Table 1 already has a caption: Review of Comparative Foundational Technologies. Please let us know if something else was meant.

  1. A framework is a basic structure underlying a system, concept, or project that serves as a foundation for development. But in the Subsection 6.1, I cannot see a picture to illustrate the basic domains, I/O relations, information flowing of the ICF. It’s better present it by a diagram.

Thank you for the feedback, a diagram has been added (Figure 4). In this diagram, the Smart Environment is depicted in the left, representing the physical space where the framework operates. Sensor data and user inputs flow into the Intelligent Connectivity Framework (ICF), which comprises various components such as IoT/IoMT, AI/ML, Building Management Systems (BMS), Healthcare Automation Systems (HAS), and Cyber-Physical Systems (CPS).

The ICF processes the inputs, leveraging the capabilities of these components, and generates outputs such as automated actions, healthcare decisions, and data analytics. These outputs ultimately contribute to improved healthcare delivery and patient outcomes, as illustrated in the right side of the diagram.

This diagram provides a visual representation of the ICF, highlighting its key components and their relationships, as well as the input-output flow of information and actions within the framework.

  1. In the Subsection 6.5, there is no real proof of concept. There is no example, no experiment, no prototype, so you did not prove your concept. You only give a basic steps, so it is still in your concept.

  1. A feasibility study is an analysis that examines the practicality and viability of a proposed project or system. It aims to assess whether the project is technically feasible, economically viable, and operationally achievable. In the Subsection 6.6, obviously, I cannot any conclusions of the so called feasibility. What you have discussed are also the contents or basic requirements.

Thank you for raising these two valid points. We have revised the titles for subsections 6.5 and 6.6 and added “Steps towards…”as well as the first paragraph of subsection 6.6 to reiterate that these are not the actual “proof-of-concept” and “feasibility study” rather than basic steps towards them.

We have also included a line (line 867) to emphasize the category of the paper as being a review paper, therefore any proposed research solution presented is of basic and introductory nature.

  1. The contents of the Subsection 7.3 is better to combine with the Figure 1.

As suggested by the reviewer, Figure 1 (reused and shown as a new figure: figure 5) was extended to display the inputs and outputs of the digital-twin ready model.

  1. In the Figure 4, what is Health 4.0? Please give a clear definition.

An entire paragraph was added (line 1116) to define what Health 4.0 is in Figure 6 (which was Figure 5 in the previous version of the paper) and how it relates to the objectives of the paper.

Reviewer 3 Report

Comments and Suggestions for Authors

Thank you for the opportunity to review this engaging paper. After careful consideration, I am pleased to observe that the revisions have notably strengthened the manuscript. A particular strength of the paper is its insightful exploration of digital health integration within smart environments. The proposed solution framework effectively addresses healthcare delivery needs and user requirements, incorporating wearable health devices, remote patient monitoring, AI-driven health analytics, and Digital Twin Technology.

Author Response

Dear Reviewer, the authors thank you for the time for the second review.

Reviewer 4 Report

Comments and Suggestions for Authors

The revised manuscript "Enhancing Healthcare through Sensor-Enabled Digital Twins in Smart Environments: A Comprehensive Analysis" shows notable improvements, particularly in the clarity of the abstract and methodology. The authors have effectively addressed previous concerns, making the paper more informative and accessible. However, future iterations could benefit from deeper exploration into user experience across diverse demographics and further refinement in balancing technical detail with readability. With these considerations in mind, and recognizing the manuscript's contribution to the field, I recommend it for acceptance, while encouraging ongoing refinement in future work.

Comments on the Quality of English Language

Moderate editing of English language required

Author Response

(The authors gave the same response as above.)

Round 3

Reviewer 2 Report

Comments and Suggestions for Authors

The authors provided a response to the comments.

Comments on the Quality of English Language

Good